# Layered subsurface in Utopia Basin of Mars revealed by Zhurong rover radar

Chao Li[1,8], Yikang Zheng[2,8], Xin Wang[1,8], Jinhai Zhang[1,8], Yibo Wang[2,3], Ling Chen[3,4✉], Lei Zhang[1], Pan Zhao[4], Yike Liu[2], Wenmin Lv[1], Yang Liu[5,6], Xu Zhao[1], Jinlai Hao[1], Weijia Sun[1], Xiaofeng Liu[7], Bojun Jia[7], Juan Li[1,3], Haiqiang Lan[4], Wenzhe Fa[7], Yongxin Pan[1,3] & Fuyuan Wu[3,4]

Exploring the subsurface structure and stratification of Mars advances our understanding of Martian geology, hydrological evolution and palaeoclimatic changes, and has been a main task for past and continuing Mars exploration missions[1–10]. Utopia Planitia, the smooth plains of volcanic and sedimentary strata that infilled the Utopia impact crater, has been a prime target for such exploration as it is inferred to have hosted an ancient ocean on Mars[11–13]. However, 45 years have passed since Viking-2 provided ground-based detection results. Here we report an in situ ground-penetrating radar survey of Martian subsurface structure in a southern marginal area of Utopia Planitia conducted by the Zhurong rover of the Tianwen-1 mission. A detailed subsurface image profile is constructed along the roughly 1,171 m traverse of the rover, showing an approximately 70-m-thick, multi-layered structure below a less than 10-m-thick regolith. Although alternative models deserve further scrutiny, the new radar image suggests the occurrence of episodic hydraulic flooding sedimentation that is interpreted to represent the basin infilling of Utopia Planitia during the Late Hesperian to Amazonian. While no direct evidence for the existence of liquid water was found within the radar detection depth range, we cannot rule out the presence of saline ice in the subsurface of the landing area.

Subsurface stratification on Mars preserves key records to decipher the geological evolution, the hydrological cycle and the palaeoclimatic and palaeoenvironmental changes of the planet[1–8]. One efficient tool to investigate the shallow structure of a planet is ground-penetrating radar (GPR) equipped on an exploration rover. As exemplified by recent studies on the Moon[14–17], GPR is capable of imaging the subsurface up to a depth of several hundreds of metres with metre-scale resolution. On Mars, rover GPR is available from two continuing missions, Perseverance[9] and Tianwen-1 (ref. [10]). Such missions with roving GPR capability aim to probe the detailed subsurface structure of the landing areas and to establish the geological framework, as well as find critical components that may constitute a habitable environment on Mars, either presently or in its past.

On 15 May 2021, China's first Mars mission, Tianwen-1, successfully deployed the Zhurong rover in southern Utopia Planitia, a topographically transitional region between the southern highlands and the northern lowlands of the Martian crustal dichotomy[11] (Fig. 1a). The landing area is mapped as the Late Hesperian lowland unit[18] (Fig. 1b), constituting predominantly the Vastitas Borealis Formation (VBF)[19] that was formed as flood-related outflow-channel sediments reworked by near-surface, volatile-driven processes[13] or as a sublimation residue of a vanished ocean[12]. Not only the origin but also the stratification of the VBF, as well as

its postformation history, at present lack observational constraints. These ambiguities hinder a deep understanding of the sedimentation history and associated geological processes of the northern lowlands of Mars.

Utopia Planitia is also characterized by distinctive geomorphic features including giant polygons, pitted cones and layered-ejecta craters that indicate a large body of water/ice might have existed there in the past[20–25] (Fig. 1c). However, it remains uncertain the extent to which hydraulic sediments are found in the subsurface and whether or not water is still present at depth in this region. East of Utopia Planitia, the volcanic eruption of Elysium Mons (Fig. 1a) during the Late Hesperian to Amazonian resulted in volcanic flows and associated debris flow deposits overlying the central and southeastern Utopia Basin[26–28], thus representing an episode of widespread resurfacing in Utopia Planitia. However, because of possible subsequent reworking, it is unclear whether or not the volcanic flows of the Elysium eruption or unrecognized late-stage volcanism on Mars has affected the vast plains far from Elysium Mons including the Zhurong landing area where the closest volcanic outcrops are located several hundreds of kilometres to the north (Fig. 1b). Recent geomorphological and chronological studies of the Zhurong landing site suggest that resurfacing probably occurred in this area during the Middle to Late Amazonian epochs[23,25], but the nature of such resurfacing events has been poorly constrained.

[1]Key Laboratory of Earth and Planetary Physics, Institute of Geology and Geophysics, Chinese Academy of Sciences, Beijing, China. [2]Key Laboratory of Petroleum Resource Research, Institute of Geology and Geophysics, Chinese Academy of Sciences, Beijing, China. [3]College of Earth and Planetary Sciences, University of Chinese Academy of Sciences, Beijing, China. [4]State Key Laboratory of Lithospheric Evolution, Institute of Geology and Geophysics, Chinese Academy of Sciences, Beijing, China. [5]State Key Laboratory of Space Weather, National Space Science Center, Chinese Academy of Sciences, Beijing, China. [6]Center for Excellence in Comparative Planetology, Chinese Academy of Sciences, Hefei, China. [7]Institute of Remote Sensing and Geographical Information System, School of Earth and Space Sciences, Peking University, Beijing, China. [8]These authors contributed equally: Chao Li, Yikang Zheng, Xin Wang, Jinhai Zhang. ✉e-mail: lchen@mail.iggcas.ac.cn

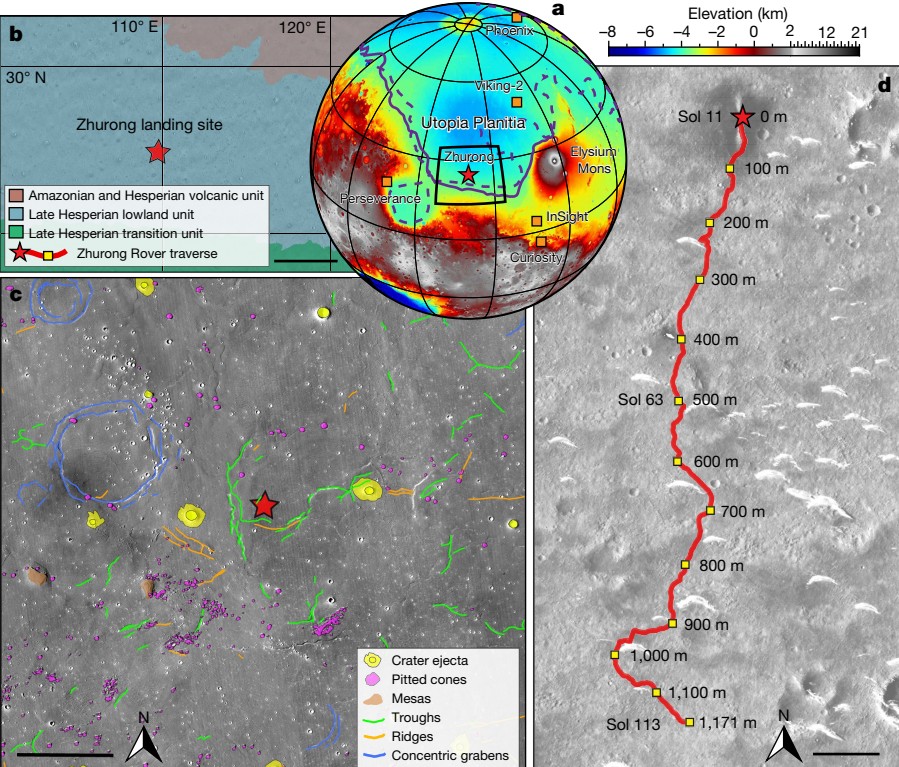

**Fig. 1 | Region around the Zhurong rover landing site. a**, Topographic map showing the landing site of Zhurong (red star), as well as the landing sites of the Phoenix, InSight, Curiosity, Perseverance and Viking-2 landers/rovers (orange squares). The purple solid and dashed lines show the locations of Martian palaeoshorelines of ref. [11], roughly delineating the depositional contact of the VBF in the northern plains. **b**, Simplified geological map near the Zhurong landing site with data from ref. [18]. Scale bar, 200 km. **c**, Geomorphic map of the Zhurong landing area with data from ref. [22]. Scale bar, 15 km. **d**, Traverse of the Zhurong rover from 25 May (Sol 11) to 6 September (Sol 113) 2021 on the basemap of a Tianwen-1 High Resolution Imaging Camera image (Sol 19, 2 June 2021). The red star marks the landing site (25.066° N, 109.925° E) and the red line shows the track of the rover. Scale bar, 100 m. Relative distances to the landing site are marked alongside the track.

## Multi-layered subsurface structure

The GPR onboard the Zhurong rover, the Rover Penetrating Radar (RoPeR), is equipped with a high-frequency channel (450–2,150 MHz) and a low-frequency channel (15–95 MHz), capable of penetrating 3–10 m and up to roughly 100 m, respectively, below the Martian surface, depending on the dielectric properties of subsurface materials[10]. Between 25 May (Sol 11) and 6 September 2021 (Sol 113), RoPeR acquired radar-sounding data over a distance of roughly 1,171 m with an approximately 8 m increase in elevation southwards from the landing site (Figs. 1d and 2a). In this study, we used the data from the RoPeR low-frequency channel to image with unprecedented high resolution the subsurface structure down to roughly 80 m depth along the traverse of the Zhurong rover, thus providing observational constraints for understanding the sedimentary history and hydrological evolution of Utopia Planitia.

We analysed the data from the RoPeR low-frequency channel (Extended Data Fig. 1) and constructed the radar reflection profile with a series of processing procedures, including preprocessing, noise attenuation, migration and topographic correction with optimal parameters (Methods and Extended Data Figs. 2–5). The resultant radar profile shows depth-varying reflection characteristics within the depth range of 10–80 m (Fig. 2a and Extended Data Fig. 5), which is the focus of our analysis and interpretation. We also estimated the dielectric permittivity (without considering dielectric loss) in the depth range of 0–80 m on the basis of diffraction analysis (Methods and Extended Data Fig. 6).

According to the pattern of reflection characteristics and the estimates of dielectric permittivity, we divide the subsurface structure into four layers (Fig. 2a,b). The first layer is no thicker than 10 m, with an average

dielectric permittivity ranging from 3–4 (Fig. 2c). However, the top part of the low-frequency radar profile is highly contaminated by strong artefacts (Fig. 2a and Extended Data Fig. 5), probably associated with multiple reflections between the rover and the ground surface. Thus, it is hard to determine the depth of the base of this top layer and separate it from the underlying materials. The second layer, extending from 10 to 30 m, in which the radar reflections are discontinuous and distributed unevenly (that is, matrix supported), nonetheless shows a general weak-to-strong change with depth accompanied by an increase in average dielectric permittivity to 4–6 (Fig. 2c). No sharp interface is observed within this layer (Fig. 2a). These features suggest that the second layer contains rocky blocks, the clast sizes of which increase with depth.

The third layer is within the depth range of 30–80 m, which has a similar weak-to-strong reflection variation pattern to the second layer, but with stronger reflections and higher values of average dielectric permittivity (ranging from 6–7, Fig. 2c), suggesting that there are larger rocky blocks distributed more evenly (that is, clast supported) at greater depth than in the overlying layer. The third layer also shows no obvious interface within it, indicating a relatively gradual change in clast size with depth between the upper and the lower parts of the layer. The bottom of this third layer is not imaged without ambiguity, either because there is no sharp stratigraphic contact or the energy of radar reflections gradually decays at depths of roughly 70–80 m (Fig. 2a). In the basal layer below approximately 80 m, the fourth layer, radar reflections are too weak and too diffuse to identify any coherent structure, suggesting that this layer is either out of the detection range of the low-frequency channel or characterized by weak internal reflections. Such ambiguity precludes further interpretation of the

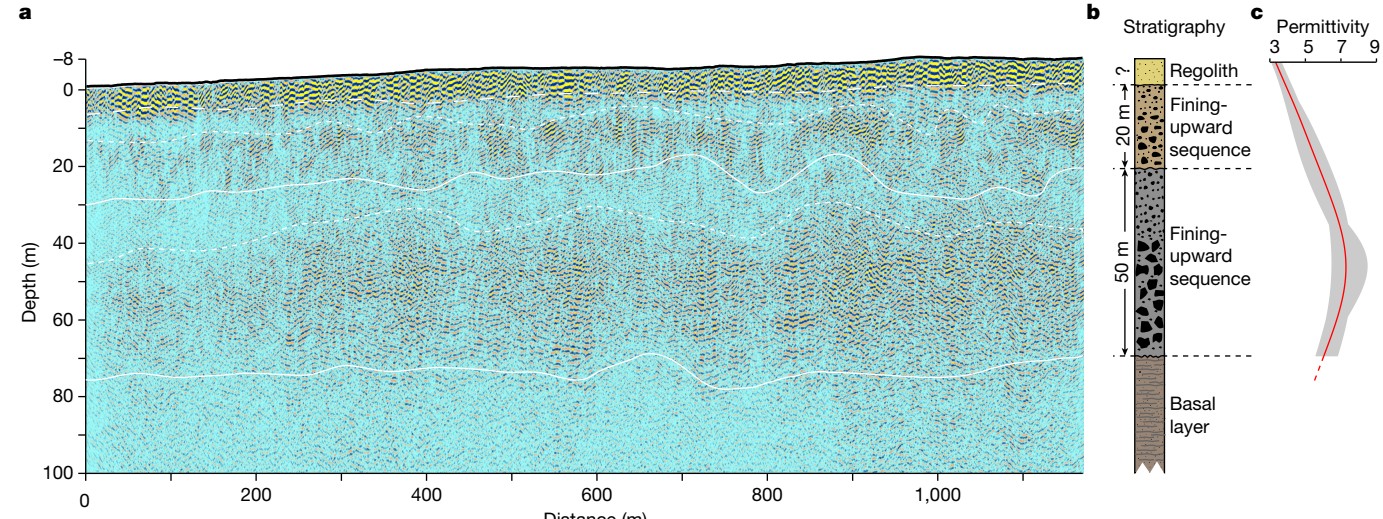

**Fig. 2 | Imaging result and interpretation of the low-frequency radar data. a**, The low-frequency radar imaging profile, with the uppermost thick black line denoting the topography relative to the landing site. The dashed line above 10 m denotes the estimated bottom of the top layer presumably containing mainly regolith. The two solid lines at depths of around 30 and 80 m represent the contacts between the second and third layers and the base of the third layer, respectively. The two dashed lines at around 10 and 40 m deep roughly separate finer- and coarser-grained rocky blocks within the second and third layers, respectively. **b**, The interpreted lithologic stratigraphy based on radar imaging. **c**, The variation of dielectric permittivity with depth. The red line is the averaged 1D dielectric permittivity profile and the bounding grey band denotes the variations around the average dielectric permittivity at each depth. Dielectric permittivity below roughly 80 m is not well constrained (see text for details).

basal layer. Our numerical simulation results of the expected radar response of rocky blocks of various sizes are broadly consistent with the observed low-frequency data (Extended Data Figs. 7 and 8), which supports the stratigraphic interpretation of the radar reflections in terms of both the variations in grain/clast size and the spatial distribution of rocky blocks.

## Basin infilling and resurfacing

The data from the RoPeR low-frequency channel show a multi-layered subsurface structure beneath the Zhurong landing area in southern Utopia Planitia, the first of its kind identified on Mars. The uppermost layer with a thickness of less than 10 m is interpreted as the Martian regolith. The second and third layers are taken to represent two fining-upwards sequences. The upper sequence is roughly 20 m thick and probably constitutes small boulders and cobbles in its lower portion. The lower sequence is much thicker, up to around 50 m thick, and the observed enhanced radar reflections, in combination with the synthetic modelling results, indicate the existence of metre-scale boulders in its lower part (Fig. 2 and Extended Data Figs. 7 and 8).

In addition to this layering, an important structural feature is the smooth transitions between layers (Fig. 2a and Extended Data Fig. 8a,b), in contrast to the subsurface structure at the InSight landing site in western Elysium Planitia where sharp interfaces are imaged in between basaltic and sedimentary strata[29]. Such a difference, combined with the geological and geomorphic observations (Fig. 1b), indicates that, in the top 80 m beneath the Zhurong landing area, any intact layer of consolidated lava flows from the Elysium eruption or unrecognized late-stage volcanism may have been either absent or too thin to survive subsequent reworking. Otherwise, there would be a strong reflection interface at the base of this layer resulting from the large dielectric contrast between basalt and sedimentary rock. This interpretation is also supported by the average dielectric permittivity of 3–7 (Fig. 2c), which differs greatly from that of the Amazonian Elysium volcanic unit (around 9), but agrees with the value of the VBF (around 5) estimated by the Mars Advanced Radar for Subsurface and Ionosphere Sounding within the similar depth range[30].

The thin upper sequence at depths of 10–30 m (Fig. 2) could reflect Amazonian resurfacing in the Zhurong landing area[23,25]. On the basis of the crater diameter-rim height relation for fresh craters[31], a reduction in the crater population with diameters less than 1.1 km because of a Middle Amazonian resurfacing event at around 1.6 billion years ago (Ga)[23] corresponds to a thickness of roughly 40 m for the infilling materials of the craters, suggesting that the Middle Amazonian resurfacing could account for subsurface materials to more than 30 m deep. Thus, it is reasonable to consider the upper sequence to be the result of resurfacing since around 1.6 Ga. Long-term weathering and repeated impacts are two surface processes possibly involved in the Amazonian resurfacing and potentially responsible for the upper fining-upwards sequence[32,33] (Fig. 3b). Either of these two processes, or a combination, have been proposed for the formation of similar near-surface fining-upwards sequences on Mars[34] and the Moon[17]. Alternatively, aqueous processes involving sedimentation to account for the Amazonian resurfacing (Fig. 3b) also need consideration. In the Zhurong landing area, the widespread presence of Amazonian-aged layered-ejecta craters[35] and pitted cones that potentially have a mud-volcano origin[24] (Fig. 1c) indicate the possible occurrence of cryosphere-fracturing-induced transient floods, especially during a period of high obliquity in the Amazonian[36,37], which may have led to the deposition of the upper sequence.

The thicker lower sequence in the depth range of 30–80 m (Fig. 2) may represent an older, probably more substantial resurfacing event of the Zhurong landing site. This resurfacing could be Late Hesperian–Early Amazonian in age, given the consistent 3.5–3.2 Ga ages from the crater size–frequency distribution over various spatial ranges in southern Utopia Planitia[21,23,24]. Both the predominance of the Late Hesperian VBF around the landing area[13,19,21] and the relatively low dielectric permittivity (Fig. 2c) similar to that of the VBF[30] suggest that the lower sequence may represents an upper portion of the VBF deposits, which could have a thickness of up to roughly 270 m at the landing area[23]. In this scenario, the fining-upwards nature of the sequence indicates that the deposition of the VBF may have been related to the rapid catastrophic flooding of southern Utopia Planitia[21] (Fig. 3a).

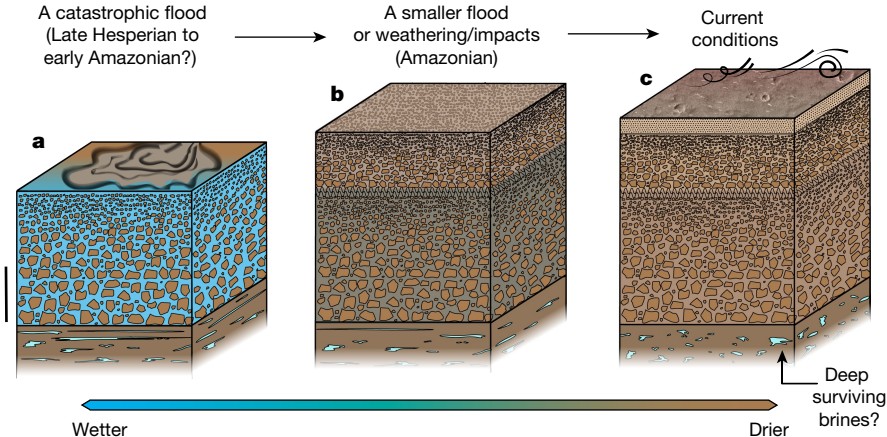

A catastrophic flood (Late Hesperian to early Amazonian?) → A smaller flood or weathering/impacts (Amazonian) → Current conditions

Wetter — Drier

Deep surviving brines?

**Fig. 3 | Conceptual model of the ancient resurfacing of southern Utopia Planitia of Mars. a**, During the Late Hesperian to Early Amazonian a catastrophic flood event occurred, leading to the formation of a fining-upwards sequence of conglomerate deposits as the flood discharge subsided, corresponding to the upper VBF. Scale bar, 20 m. **b**, A resurfacing event probably associated with a transient flood, or long-term reworking by weathering or repeated impacts, occurred in the Amazonian, resulting in a fining-upwards sequence with relatively smaller boulders and cobbles atop the consolidated older sediments. **c**, The subsequent loss of water to high latitudes resulting from the modern high obliquity of Mars led to the formation of the dry near-surface regolith and dominantly aeolian deposition/erosion processes at present.

## Possible existence of subsurface ice

One of the primary goals of RoPeR is to probe whether there is subsurface water/ice in southern Utopia Planitia, particularly as distinctive geomorphic features in this region suggest that a substantial amount of water/ice might have existed in the geologic past. Our low-frequency radar imaging profile shows radar signals within the depth range of 0–80 m (Fig. 2a), precluding the existence of a water-rich layer within this depth range as the existence of water would strongly attenuate the radar signals and diminish the visibility of deeper reflections. The estimated low (less than 9) dielectric permittivity (Fig. 2c) further supports the absence of a water-rich layer as water-bearing materials typically have high (greater than 15) dielectric permittivity[7]. We further tested this assessment with thermal considerations by conducting a heat conduction simulation based on available thermal parameters estimated from previous studies (Methods). Our thermal simulation results (Extended Data Fig. 9d) show that the Zhurong landing area has an annual average temperature of around 220 K in the RoPeR detection depth range, which is much lower than the freezing point of pure water (273 K), and also lower than the eutectic temperatures of typical sulfate and carbonate brines, but slightly above those of perchlorate brine systems[38]. This observation suggests that the shallow subsurface of the Zhurong landing area could not stably contain liquid water nor sulfate or carbonate brines, consistent with the radar imaging result.

The combination of our temperature estimates (Extended Data Fig. 9) and radar image (Fig. 2a) suggests that the presence of perchlorate brine is possible, but might only occur deeper than roughly 80 m. At this stage, we cannot rule out the existence of saline ice in the presence of sulfate or carbonate, as the dielectric permittivity of these materials (2.5–8) is indistinguishable from rocky materials (Fig. 2c). Liquid water is proposed to exist under the polar ice caps of Mars[7,39], and water ice is also reported to be present at shallow depths of low-to-mid-latitude regions[40–42]. Our results from southern Utopia Planitia do not provide evidence for the presence of water in the upper roughly 80 m. Liquid water and/or brines, if they exist, may have been buried at greater depths (Fig. 3c), mostly beyond the penetrating depth of RoPeR.

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

# Article

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

# Methods

## RoPeR data processing

The RoPeR is equipped with two channels, including a high-frequency channel with an operating frequency range of 450–2,150 MHz and a low-frequency channel with a bandwidth ranging from 15 to 95 MHz. Note that the so-called high-frequency channel and low-frequency channel were defined specifically for the RoPeR onboard the Zhurong rover[10], and therefore the same terms are used here. The high-frequency channel has a penetrating depth of about 3–10 m with a vertical resolution of a few centimetres, and the low-frequency channel can penetrate down to roughly 80 m deep with a metre-scale vertical resolution, depending on the dielectric properties of subsurface materials[10]. In this study, we processed the data from the low-frequency channel of RoPeR (Extended Data Fig. 1) to image the subsurface structure. The data processing for the RoPeR low-frequency channel consists of the following steps.

(1) Self-test trace removal. The self-test traces from the RoPeR low-frequency channel are used specifically for checking the status of the RoPeR module. These traces therefore contain no effective subsurface information and should be excluded before processing. All the self-test traces were identified by the character string 'Self_Test' in the corresponding data label file and then removed from the raw data. The RoPeR low-frequency channel has a total of 2,863 traces after removing self-test traces (Extended Data Fig. 2a).

(2) Trace-spacing regularization. The low-frequency channel of RoPeR took measurements every 25–50 cm, depending on the operation parameter. After 17 August 2021, the trace spacing of the low-frequency channel changed from 50 to 25 cm. We regularized the trace spacing by downsampling the data after 17 August 2021. The regularized data have 2,289 traces with an average trace spacing of 50 cm (Extended Data Fig. 2b).

(3) Direct current shift removal. The direct current shift was estimated as the average of the samples before time zero and was subtracted from each trace (Extended Data Fig. 2c).

(4) Time zero correction. The time zero of the low-frequency channel is 212.5 ns according to the Ground Research and Application System (GRAS) of China's Lunar and Planetary Exploration Programme. Thus, the data before time zero were removed[10] (Extended Data Fig. 2c).

(5) Background removal. This step, essentially subtraction of the mean related to background, was achieved by subtracting the average value of each segment of the traces (Extended Data Fig. 3a).

(6) Band-pass filtering. On the basis of the operating frequencies of low-frequency channel[10], we applied band-pass filtering between 15 and 95 MHz to enhance the signal-to-noise ratio. This step, however, leads to less noticeable changes in the data compared with the previous step.

(7) Automatic gain control. With increasing propagation depth, the energy of radar echoes decreases gradually. Thus, we systematically applied automatic gain control to boost the energy of radar echoes from deep reflectors (Extended Data Fig. 3b).

(8) Random noise attenuation. Random noise was suppressed to enhance the visibility of the whole profile[43] (Extended Data Fig. 3c). We adopted a streaming orthogonal prediction filtering method for denoising, which has proved to be capable of effectively eliminating random noise while preserving real signals[43].

(9) Migration. Using the velocity model estimated by diffraction separation and focusing analysis (see Dielectric permittivity estimation by diffraction analysis), the radar profile was migrated and further converted from the time domain to the depth domain to recover both shapes and depths of reflectors[16,17] (Extended Data Fig. 4).

(10) Topographic correction. We corrected the topography of the migrated radar profile using the relative elevation information provided by GRAS (Extended Data Fig. 5).

## Dielectric permittivity estimation by diffraction analysis

Diffractions are echoes from small-scale subsurface anomalies, such as small blocks and fractures, which carry abundant subsurface velocity information that is crucial for estimating the dielectric permittivity for GPR[44–47]. First, we used the plane-wave destruction method[48,49] to separate diffractions from reflections. Then, we used those separated diffractions to construct the subsurface macro-velocity model by focusing analysis[44,45]. Finally, we converted the velocity model into a dielectric permittivity model using the following equation:

$$\varepsilon_r = \left(\frac{c}{v}\right)^2,$$

where $\varepsilon_r$ is the dielectric permittivity, $c$ is the speed of light in vacuum and $v$ is the subsurface velocity.

As shown in Extended Data Fig. 6, from 0 to 80 m depth, there are mainly three layers according to the values of dielectric permittivity, consistent with the reflection pattern shown in Fig. 2a in the main text. Beneath around 80 m deep, the dielectric permittivity is not well constrained because the number of effective radar echoes is insufficient. To better illustrate the depth-dependence of dielectric permittivity, we derived the averaged one-dimensional (1D) dielectric permittivity profile (Fig. 2c) from the two-dimensional (2D) dielectric permittivity image.

## Numerical simulation of GPR

We performed numerical simulation of the GPR on the Zhurong rover to verify the validity of our stratigraphic interpretation. According to the radar imaging results (Fig. 2), we designed a numerical model of dielectric permittivity by assuming different layers containing rock clasts of varying size and abundance in a sandy matrix (Extended Data Fig. 7). Such a method has been applied previously for verifying the imaging results of lunar penetrating radar data[16,17,50,51]. In our interpretation of the radar profile, we focus on the reflection pattern in the depth range in which radar signals are visible. Previous radar simulation results with frequency contents comparable to the RoPeR low-frequency data show that the presence of weak dielectric losses does not noticeably affect the reflection pattern in which radar signals dominate over noise[52]. As there is little liquid water detected within the penetration depth range (0–80 m) of the RoPeR low-frequency data at the Zhurong landing site, the corresponding dielectric loss is expected to be weak and thus not to affect the simulated reflection pattern. Therefore, we neglected dielectric loss in the numerical model. We applied the finite-difference method to solve the Helmholtz equation in the numerical simulation[16,17,50,51]. We compared the simulation results with the observation and adjusted the sizes and spatial distribution of rocks (Extended Data Fig. 7) until the synthetics and the data show similar depth-dependent variations in both the reflection pattern and average strength envelope, as well as dielectric permittivity (Extended Data Fig. 8).

## Thermal simulation for the Zhurong and Phoenix landing sites

To investigate the possibility of the presence of liquid water or brine in the subsurface, we calculated the three-phase diagrams (gas–liquid–solid) of briny water under the thermal and lithostatic conditions at the Zhurong landing site. For comparison, we also conducted the heat conduction simulation for NASA's Phoenix landing site, which is located further north (roughly 68° N, Fig. 1a), where ground ice has been detected[39]. The simulation process is provided below and results are presented in Extended Data Fig. 9.

(1) Temperature estimation. In the subsurface where conduction dominates, we can get the heat conduction equation as

$$\frac{\partial T}{\partial t} = \kappa \left(\frac{\partial^2 T}{\partial^2 z}\right),$$

# Article

where $T(z,t)$ is temperature as a function of depth $z$ and time $t$, $\kappa$ is thermal diffusivity and is approximately set to $1 \times 10^{-6}$ m$^2$ s$^{-1}$ according to the thermal diffusivity of ice[53] or sandstone[54]. On the ground surface ($z = 0$), $T(0,t)$ can be described in the form of sine series expansion:

$$T(0,t) = T_0 + \sum_{i=1}^{n} A_{0i} \sin\left(\frac{2i\pi}{P}t + \varphi_{0i}\right),$$

where $P$ is the duration (that is, 1 year in our calculation), $T_0$ is the annual average ground temperature, $A_0$ is the amplitude, $\varphi_0$ is the initial phase and $i$ is the order of expansion. Thus, the heat conduction equation can be written as[53]

$$T(z,t) = T_0 + \gamma z + \sum_{i=1}^{n} A_{0i} e^{-\sqrt{\frac{i\pi}{\kappa P}}z} \sin\left(\frac{2i\pi}{P}t + \varphi_{0i} - \sqrt{\frac{i\pi}{\kappa P}}z\right),$$

where $\gamma$ is a constant to describe the thermal gradient $\partial T/\partial z$, which can be determined by Fourier's law:

$$\gamma = \frac{\partial T}{\partial z} = \frac{Q_0}{k_0},$$

where the average thermal conductivity $k_0$ of Martian subsurface up to 140 m was set to be 0.8 Wm$^{-1}$K$^{-1}$ (ref. [55]); an average heat flux ($Q_0$) of 18 mW m$^{-2}$ was selected from the present-day heat flow model of Mars[56]. The annual surface temperatures ($z = 0$) at the Zhurong and Phoenix landing sites (Extended Data Fig. 9a,b) were calculated using the Mars Climate Database[57]. Given that the RoPeR data used in this study were obtained from 25 May to 6 September 2021 (UTC), when the solar longitude $L_s$ varied from 50 to 95 and the solar day from Sol 11 to Sol 113, we specifically analysed the temperature–pressure crossplots (Extended Data Fig. 9d) at $L_s = 50$–95 (Sol 11–Sol 113) for the Zhurong landing site. As we consider the seasonal changes of temperature here, the temperature in the shallowest 1 m, where diurnal temperature change dominates, is not presented in Extended Data Fig. 9d.

(2) Lithostatic pressure calculation. Lithostatic pressure $P_L$ was calculated by $P_L = \rho g z$, where $g$ is the gravitational acceleration on Mars, which is 3.693 m s$^{-2}$. Density, $\rho$, was set to linearly increase from 1,211 to 1,500 kg m$^{-3}$ with depth on the basis of the parameters given in refs. [58,59].

(3) Three phases of water and eutectic points of brines. The triple point of water is located at 273 K, 612 Pa. The eutectic points of possible brines[38,60,61] are plotted for comparison (Extended Data Fig. 9d).

## Data availability

The Tianwen-1 data including the Mars Rover Penetrating Radar (RoPeR) data and the High Resolution Imaging Camera image used in this study are processed and produced by the GRAS of China's Lunar and Planetary Exploration Programme, and provided by CNSA at https://clpds.bao.ac.cn/web/enmanager/mars1. The dataset containing the imaging result of the RoPeR low-frequency data, the estimated 2D dielectric permittivity model and the averaged 1D dielectric permittivity profile can be accessed at the World Data Center (WDC) for Geophysics, Beijing (https://doi.org/10.12197/2022GA018). Source data are provided with this paper. Other datasets generated and analysed in this study are available from the corresponding author upon reasonable request.

## Code availability

The codes used in this study are available to interested researchers upon request.

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

**Acknowledgements** We acknowledge the China National Space Administration (CNSA), China's first Mars exploration mission (Tianwen-1) team, the GRAS, and the payload team of GPR. We are grateful to R. Mitchell for constructive comments and editing the manuscript. We thank Y. Lin, H. Lin, S. Hu, B. Zhou, Y. Su, J. Liu, S. Dai, Y. Li, F. Zhang, C. Xu, X. Liang, L. Chen and Z. Zhang for their helpful discussions during this work. This study is supported by the Institute of Geology and Geophysics, Chinese Academy of Sciences (grant no. IGGCAS-202102), the Key Research Programme of the Chinese Academy of Sciences (grant no. ZDBS-SSW-TLC001) and the National Natural Science Foundation of China (grants 42288201 and 41941002).

**Author contributions** L.C. and J.Z. designed the research. C.L., Y.Z., X.W., J.Z., Y.W., L.Z., Y.L., W.L., X.Z., J.H., W.S., X.L., B.J., J.L., H.L., W.F. and Y.P. conducted the data analysis. L.Z., L.C. and J.Z. performed the thermal simulation. W.L., J.Z., L.Z., Y.Z. and Y.W. performed the numerical simulation of the GPR data. L.C., P.Z., J.Z., X.W., Y.L., Y.P. and F.W. contributed to the interpretation of the observations. L.C., X.W., P.Z., J.Z., C.L. and L.Z. wrote the manuscript. All coauthors discussed the results and commented on the manuscript.

**Competing interests** The authors declare no competing interests.

**Additional information**
**Correspondence and requests for materials** should be addressed to Ling Chen.

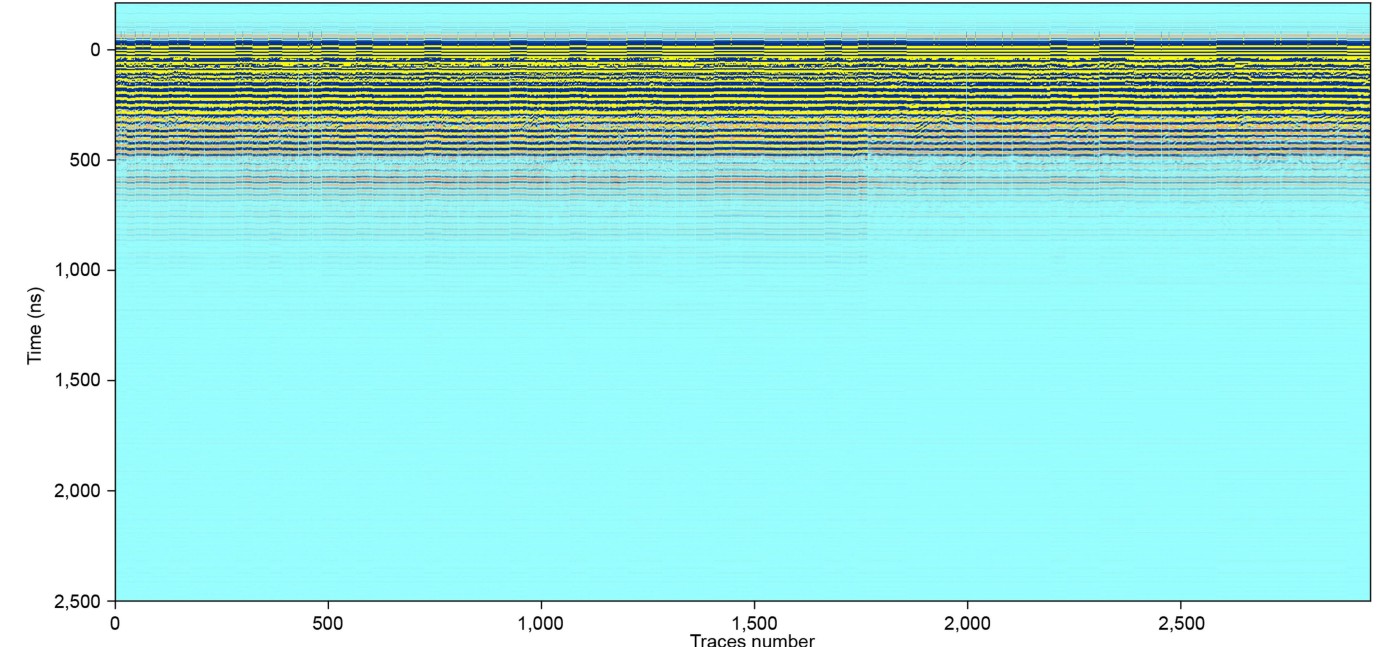

**Extended Data Fig. 1 | Low-frequency radar profile of the raw data.** The raw data have a total of 2,945 traces, including self-test traces.

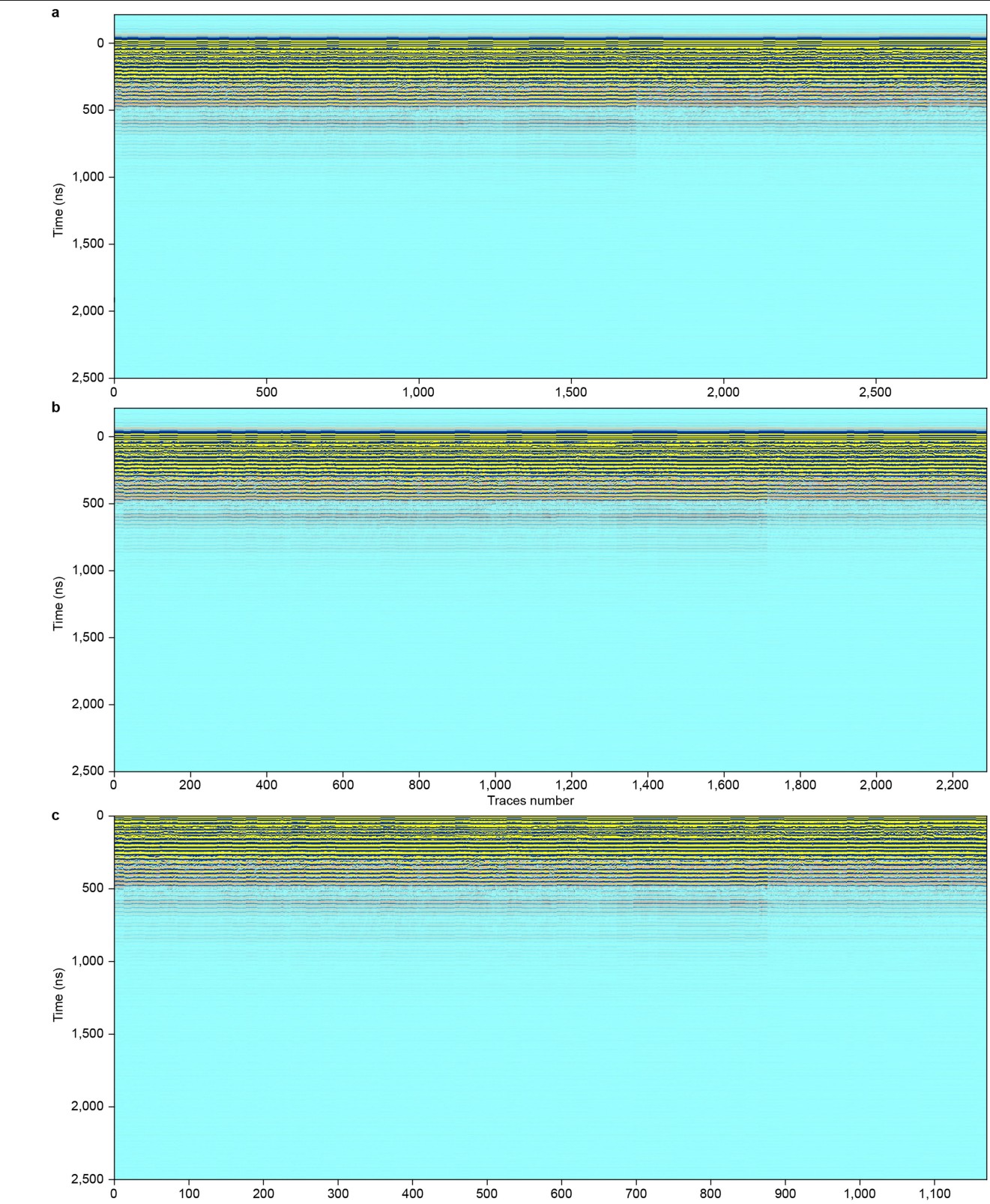

**Extended Data Fig. 2 | Low-frequency radar profiles after applying a series of data regulation processing. a**, Self-test trace removal. **b**, Trace spacing regularization. **c**, Direct-current (DC) shift removal and time zero correction.

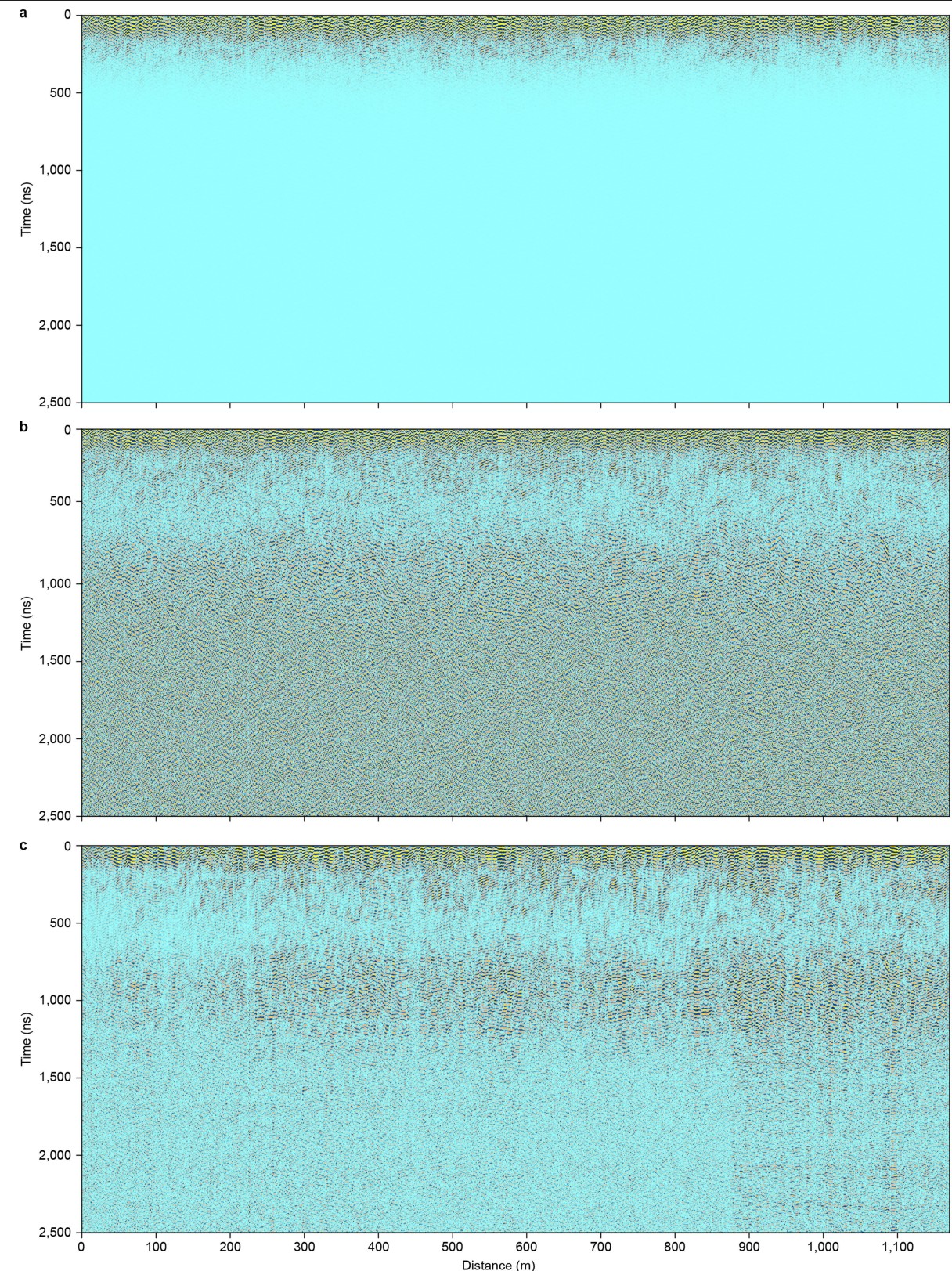

**Extended Data Fig. 3 | Low-frequency radar profiles after applying a series of signal enhancement processing. a**, Background removal. **b**, Band-pass filtering and automatic gain control (AGC). **c**, Random noise attenuation.

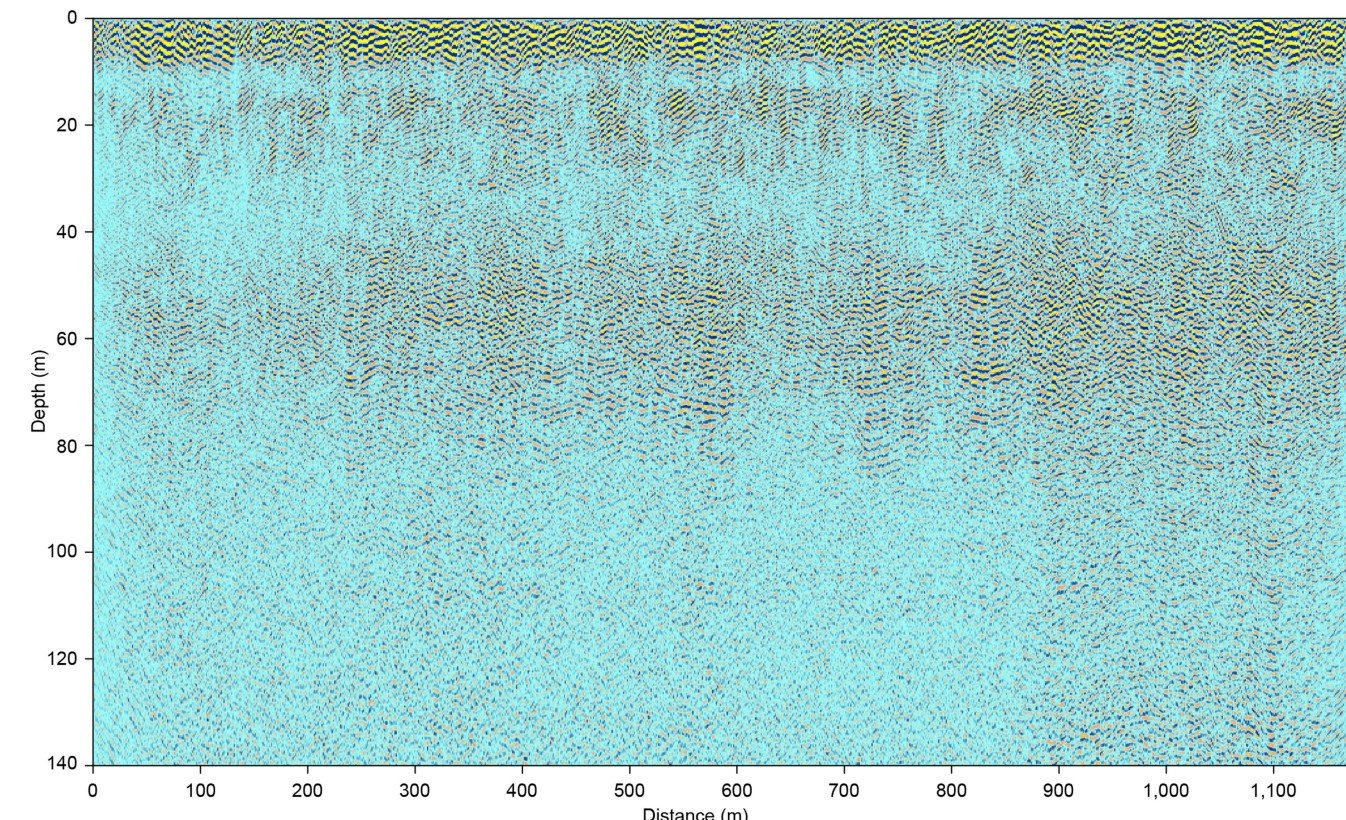

**Extended Data Fig. 4 | Low-frequency radar profile after applying migration and time-to-depth conversion.** The velocity model used was constructed by plane-wave destruction and focusing analysis.

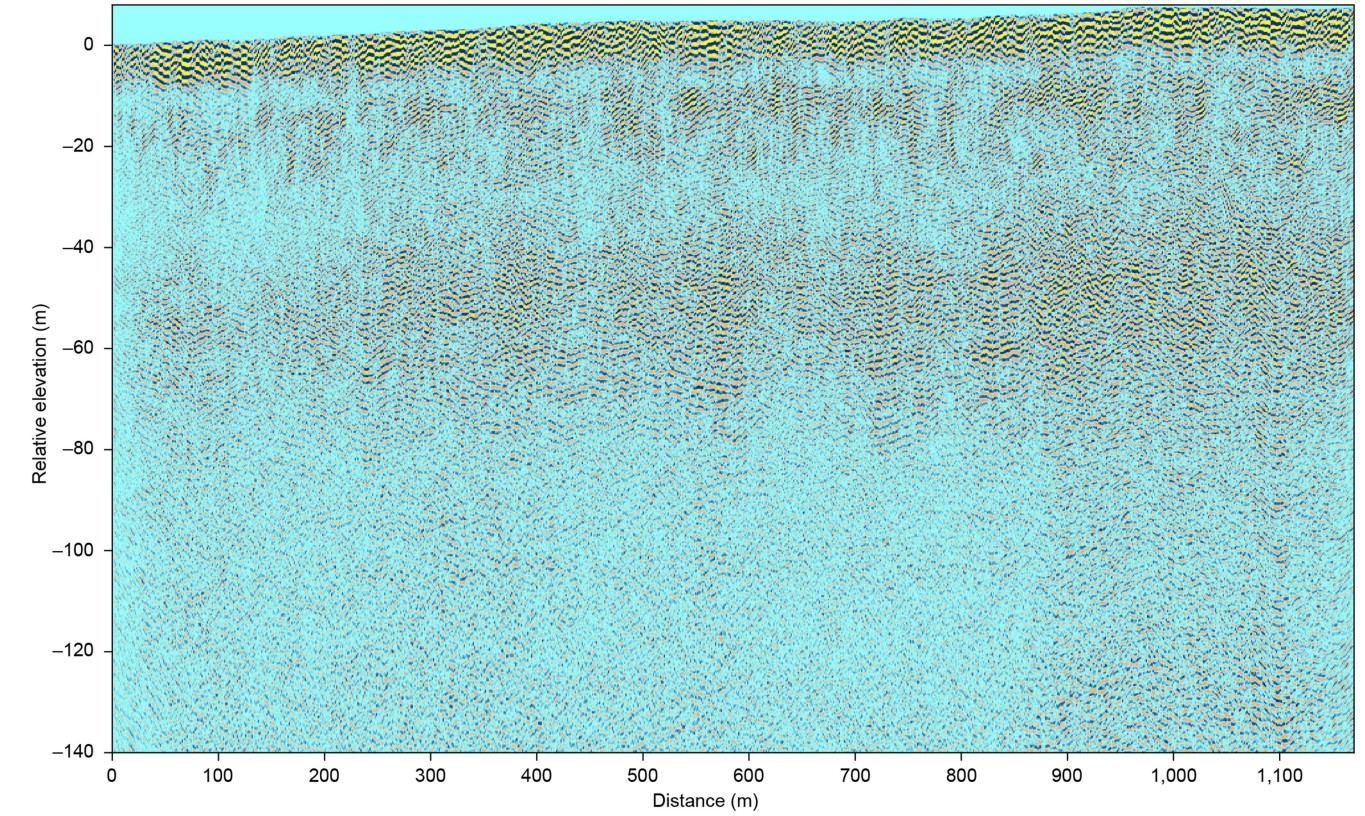

**Extended Data Fig. 5 | Low-frequency radar profile after topographic correction.** The topographic corrections range from -0.3 m to -7.9 m.

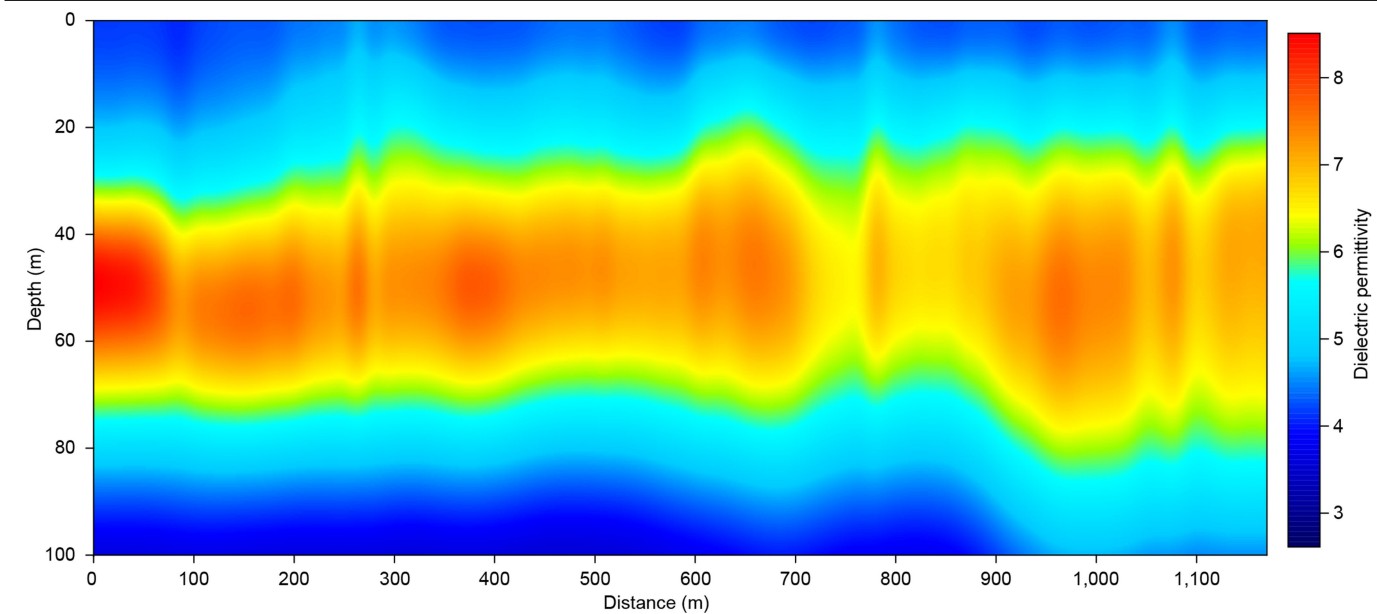

**Extended Data Fig. 6 | Dielectric permittivity model.** This model was converted from the velocity model used in migration and time-to-depth conversion.

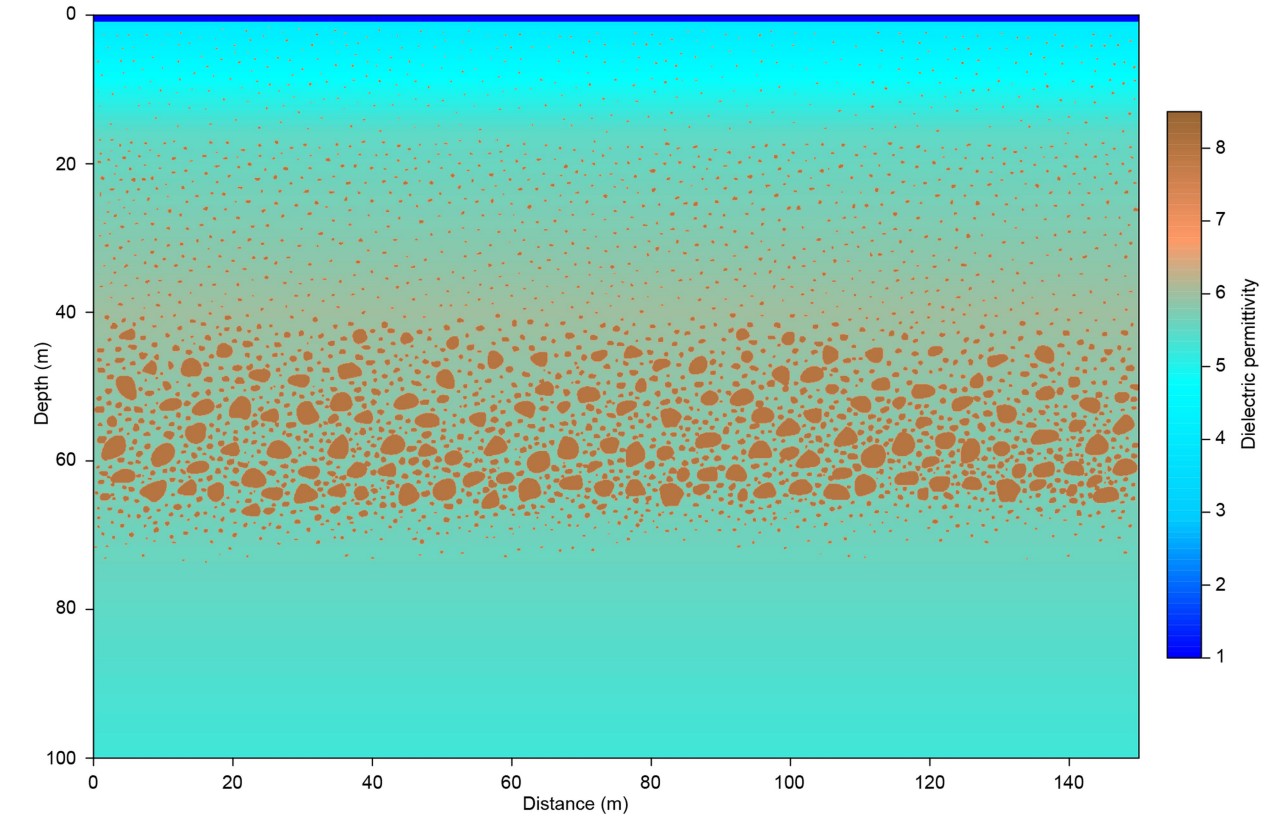

**Extended Data Fig. 7 | Numerical model of dielectric permittivity.** This model was constructed according to the stratigraphic interpretation of the radar profile shown in Fig. 2.

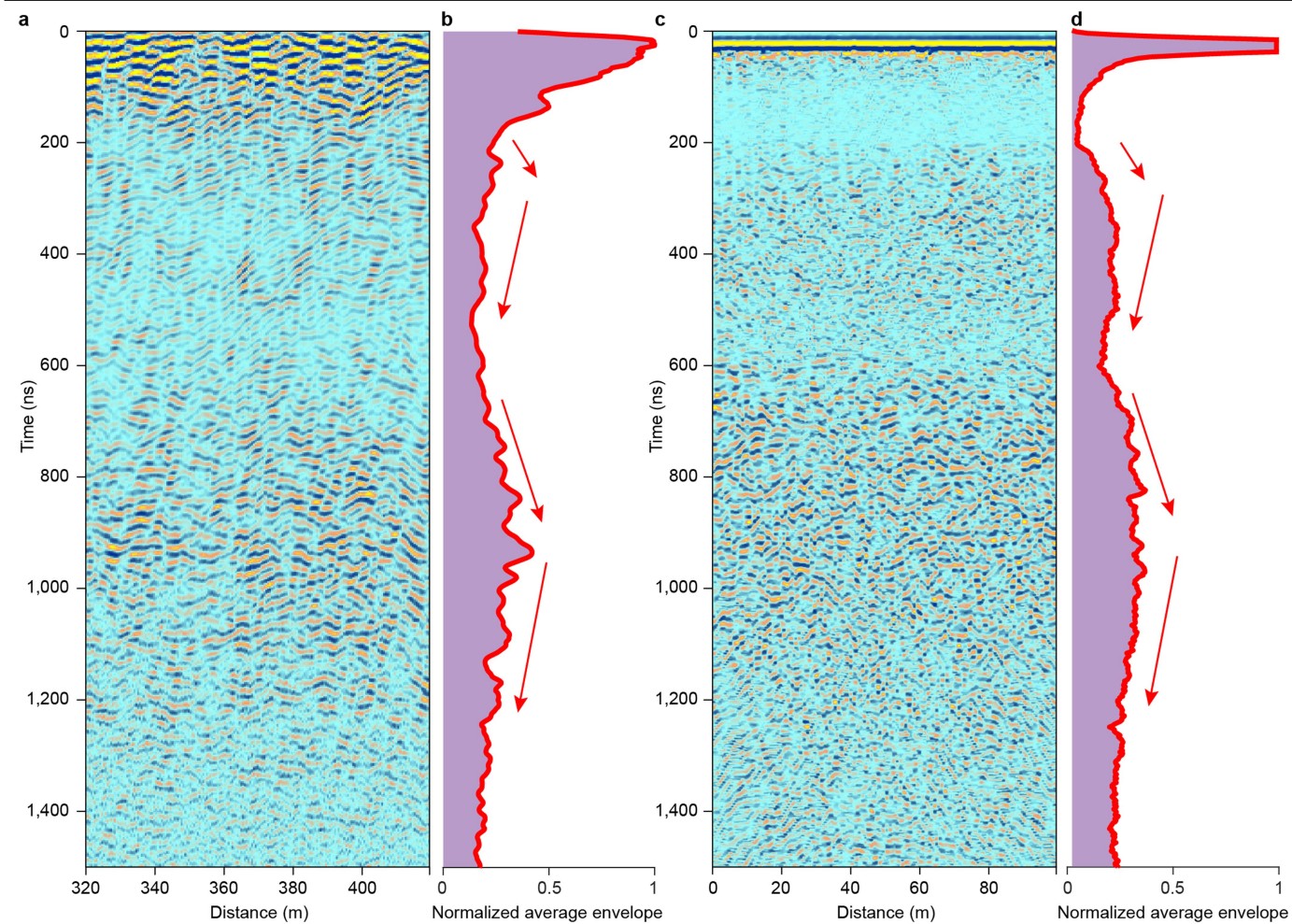

**Extended Data Fig. 8 | Comparison between the observed ground-penetrating radar (GPR) data and synthetic data. a**, Part of the Rover Penetrating Radar (RoPeR) low-frequency data (track distance from 320–420 m). **b**, The corresponding normalized average strength envelope of **a. c**, Simulated low-frequency radar data using the model shown in Extended Data Fig. 7. **d**, The corresponding normalized average strength envelope of **c**. The peak amplitudes of the strength envelope in **d** were clipped by 20% to boost the deep record. Both datasets were processed with the same processing procedures as those shown in Extended Data Figs. 2 and 3.

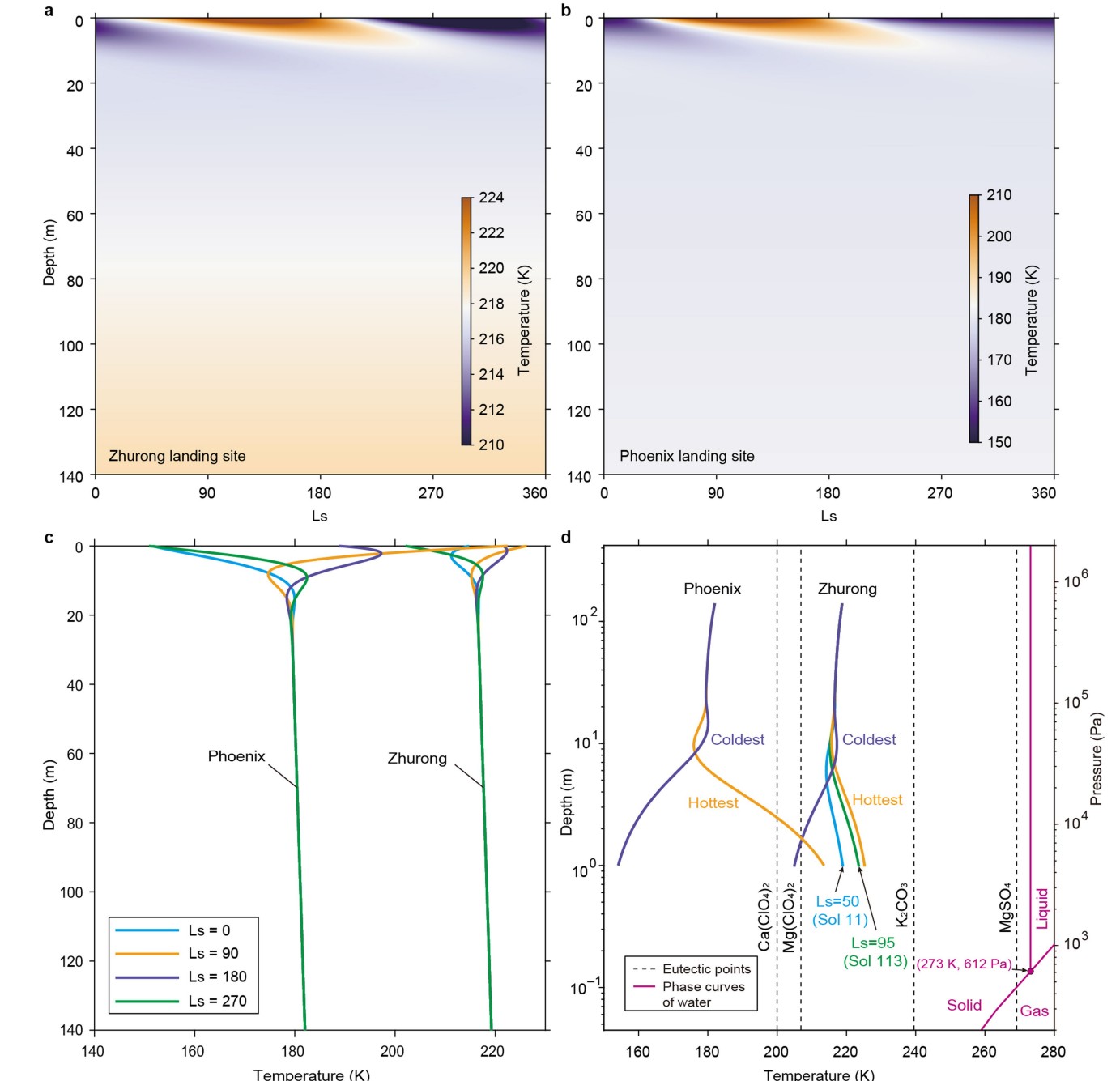

**Extended Data Fig. 9 | Thermal simulations for the Zhurong and Phoenix landing sites. a**, Temperature distribution based on the heat conduction simulation for the Zhurong landing site. **b**, The same as **a**, but for the Phoenix landing site. **c**, Temperature variation with depth, obtained from the results in **a** and **b**. **d**, Crossplots between temperature and depth/pressure for different solar longitudes (Ls). Phase curves of water (solid lines) and the eutectic points of possible brines (dashed vertical black lines) are presented. The pink dot denotes the triple point of water (273 K, 612 Pa). For the Zhurong radar data used in this paper, Ls 50 (Sol 11) and Ls 95 (Sol 113) represent the starting and the ending solar longitude (solar day), respectively.