## [Peer Review File. · Nature]

Manuscript Title: Layered subsurface in Utopia Basin of Mars revealed by Zhurong rover radar.

Editorial Note: Redactions – Third Party Material

Reviewer Comments & Author Rebuttals

Reviewer Reports on the Initial Version:

Referee #1:

In the paper "Layered subsurface in Utopia Basin of Mars revealed by Zhurong rover radar" the authors present a novel data set of ground penetrating radar at the Zhurong landing site and >1km traverse in southern Utopia Planitia. This is a very exciting data set to acquire as it may provide quantitative information about the subsurface in a portion of Mars with a poorly constrained history.

I was very excited to see this data set and believe that the novel data should be published with interpretations. However, there are some issues with the interpretations that may be ambiguous or unsubstantiated, and the data may be over-interpreted. I don't know how possible it will be to fix these issues in the Nature format.

Generally, I found the paper easy to read and the English very good. I did struggle with some technical descriptions that were left undefined or were perhaps not commonly used.

Major Issues:

One issue that can be raised relatively easily is that the Zhurong landing site is within a unit called "Late Hesperian lowland unit" - from reference 15 and Figure 1b. The Hesperian-aged interpretation is based on crater counts and has undergone peer review and community approval. Nevertheless, the manuscript contradicts this map numerous times in the interpretations and figures without any explanation - by invoking Amazonian period (younger) processes of floods, glaciers, and lava flows. If the surface were resurfaced in the relatively recent Amazonian (compared to the older Hesperian), then its age should be updated and a lot more detail should be provided, but I don't think the manuscript is trying to change the map, an otherwise natural result of detailed analysis resulting in this interpretation.

Several examples invoking Amazonian age processes to explain a Hesperian aged deposit:

- 1 Figure 3a and 3b shows "A smaller flood (Amazonian)".
- 2 Lines 38, 155, 156, 159 (middle Amazonian), 167, 186, 205, 207, etc.

For this reason alone, I believe the paper could use a major revision that considers Hesperian-aged processes rather than Amazonian-aged processes.

—

Lines 98-99, 373-374, ExFig2 I'm not familiar with techniques to suppress (or attenuate, line 99)

noise, nor am I familiar with effects of doing this, so it's incumbent on the authors to describe this process fully and transparently in order to do a good evaluation. I'm worried that suppressing noise via an un-described mechanism may introduce artifacts or a different kind of noise that cannot be as easily characterized or detected. Further, in this paper readers/reviewers cannot effects of doing "noise suppression" because this step was completed simultaneously with "background removal, band-pass filtering, [and] AGC" (Data Figure 2). This is too much processing in a single step for a good evaluation, and I am unable to assess if any of these processes introduced artifacts that can alter interpretations. This is critical to address because every interpretation of the images hinges on this step.

Minor: I'm also not familiar with the "self-test trace removal". Are these traces that are used for calibration? A citation for previous usage of this is necessary.

—

On interpretation, the manuscript presents three plausible scenarios to explain the observations and identifies one scenario that is clearly favored (Section starting in line 150; Figure 3). But I'm worried that the data are being over-interpreted by making conclusions about ambiguous data. Assuming that no artifacts were introduced in making Figure 2, without clear reflections, it may be that no unique solution can be found, meaning that there are likely more interpretations possible.

For example, In Figure 2 80-100 m, the manuscript suggests that this low reflection zone (basal layer) is caused by a gradual decrease in permittivity ($\epsilon \sim 5$) that, but it could easily arise from
1) a loss of signal before 80 m, meaning it's all noise
2) the transition at ~ 80 m is to solid bedrock with no internal reflections, leaving only noise to interpret (similar to the idea for the moon, line 196).

As I see it, the dashed line on the far right of Figure 2 is completely unconstrained and should have a very large gray band around it, reaching to >9 .

I think there is a possibility of over interpreting done at shallower depths too (see next comment)

—

Based on the previous comment and the methods (especially lines 414-424), I'm worried that the radar propagation modeling was carefully tuned to reproduce the observations and that once found, additional tuned solutions weren't sought. Radar attenuation via transmission losses (ϵ'') was not taken into account, so finding the signal strength at depth with this model leaves the possibility of other solutions being missed.

—

Minor Comments:

Line 33 "southern Utopia Planitia" is a huge region. This is at one specific site

Line 52 missing article "targeted the southern"

Lines 64-68 This feels like a distraction because those features are approximately 1000 km farther north, where the climate is very different from southern UP.

Lines 68-74 Again, a likely distraction since even Figure 1b shows that the volcanic flows didn't reach the Zhurong landing site,

Lines 77-78 It's tempting, as the manuscript does, to call these "high-frequency" and "low-frequency" channels, and for the purpose of this manuscript it's probably fine. But there is also value in acknowledging that the higher frequency channel is fully in the "Ultra high frequency" UHF part of the spectrum (300 - 3,000 MHz). In a similar way, the "low-frequency" channel is mostly

within the "Very High Frequency" VHF domain (30-300 MHz). If the authors are inclined, I'd recommend calling them the VHF channel and UHF Channel, but this is a preference and not strictly necessary.

That said, on line 97 and throughout, I recommend changing the text to say "high- and low' frequency channels." For example, "we analyzed data from both the high- and low-frequency channels and ..."

In several locations the manuscript uses strong language when there should be some hedging. For example, on line 102-103 "radar profile shows clear reflections within the depth range of 10-80 m". These aren't clear to me. Usually, "clear reflections" would be used to describe discrete reflectors that can be traced. Here's an example I found on the web:

<https://www.si.edu/newsdesk/photos/mars-sharad-data>. I don't see anything like that in Figure 2. What I see is "regions with variable reflectivity" or some qualitative description like that. Other examples include

Line 126 "indicating a graduate change in clast size" - this is an interpretation of noisy data that does not clearly indicate clast size changes.

Line 134 "independently verifies the stratigraphic interpretation" - it's hard to independently verify something with only one dataset. Normally, numerical models are used to "support the stratigraphic interpretation"

Line 163 "synthetic modeling results consistently indicate ..." very strong words.

Line 169 "observed" should be "interpreted"

Line 191 "strong reflections"

Line 218 "clear reflections" - again, they aren't clear. Because water would attenuate the signal, one could change the text to read "shows a radar signal within the depth range"

Line 147 This isn't a diffraction survey, so I'm confused by this terminology.

Lines 158-161 Big leap in logic to get to southern UP.

Line 211-212, Figure 3 This comes out of nowhere.

Line 238-240 (last sentence): This is a strange ending. I'm not sure that the interpretations are strongly suggestive that "shallow water has fully escaped" because of the ambiguities associated with the interpretations. It may be sufficient to say that this investigation has not found evidence of water in the upper 80 m.

Line 471, I'm not able to access the data without a login.

Referee #2:

The manuscript provides analysis of ground penetrating radar (GPR - RoPeR Instrument) data collected within the Utopian Basin of Mars by the Tianwen-1 - Zhurong rover. Through the collection of >1km profile, RoPeR offers a unique and important insight into the geology of the this fascinating region of the planet. The data processing steps are clearly outlined in the methodology section (and apply well established techniques from terrestrial GPR studies) and the final radargram (Fig 2/Extended Data Fig. 4) displays horizontal variations in subsurface scattering properties that are interpreted to be evidence of layering within the upper ~80m. Drawing on previous orbital data analysis the manuscript considers multiple possible geologic explanations for the RoPeR results. Ultimately, a layered sequence of sedimentary deposits laid down by episodic flooding is favored.

Based on the evidence presented I am not convinced that the RoPeR data supports the interpretation presented. The subsurface structures are not well defined in the GRP profile and

thus a wide range of processes could be responsible for the crude layering observed. To support the flood deposit interpretation, the GPR data is compared to synthetic data generated from a model of suspended boulders within a sandy matrix (Extended data fig 8). The real data displays a more complex pattern of scattering relative to the simulated data, especially between 400 and 600 ns and thus the model doesn't appear to be adequate interpretation of actual results. I therefore suggest that more modeling is performed to test the other three scenarios considered on page 9

I also appreciate that the manuscript draws on published mapping efforts to support the favored interpretation, but the resolution of the map in Fig 1 is very coarse relative to the ~1km rover profile. I suggest the authors use Tianwen-1 image data as well as HiRISE and CTX images to further investigate the landing site region to provide support for the subsurface analysis. Likewise, does the surface analysis provided by the rover provide any evidence consistent with the flood deposit hypothesis?

Author Rebuttals to Initial Comments:

Reviewer #1:

In the paper “Layered subsurface in Utopia Basin of Mars revealed by Zhurong rover radar” the authors present a novel data set of ground penetrating radar at the Zhurong landing site and >1km traverse in southern Utopia Planitia. This is a very exciting data set to acquire as it may provide quantitative information about the subsurface in a portion of Mars with a poorly constrained history.

I was very excited to see this data set and believe that the novel data should be published with interpretations. However, there are some issues with the interpretations that may be ambiguous or unsubstantiated, and the data may be over-interpreted. I don’t know how possible it will be to fix these issues in the Nature format.

Generally, I found the paper easy to read and the English very good. I did struggle with some technical descriptions that were left undefined or were perhaps not commonly used.

We appreciate the reviewer’s thoughtful review of our manuscript, and have accordingly made the following revisions to the manuscript.

Major Issues:

1. One issue that can be raised relatively easily is that the Zhurong landing site is within a unit called “Late Hesperian lowland unit” - from reference 15 and Figure 1b. The Hesperian-aged interpretation is based on crater counts and has undergone peer review and community approval. Nevertheless, the manuscript contradicts this map numerous times in the interpretations and figures without any explanation - by invoking Amazonian period (younger) processes of floods, glaciers, and lava flows. If the surface were resurfaced in the relatively recent Amazonian (compared to the older Hesperian), then its age should be updated and a lot more detail should be provided, but I don’t think the manuscript is trying to change the map, an otherwise natural result of detailed analysis resulting in this interpretation.

Several examples invoking Amazonian age processes to explain a Hesperian aged deposit:

1 Figure 3a and 3b shows “A smaller flood (Amazonian)”.

2 Lines 38, 155, 156, 159 (middle Amazonian), 167, 186, 205, 207, etc.

For this reason alone, I believe the paper could use a major revision that considers Hesperian-aged processes rather than Amazonian-aged processes.

We appreciate the reviewer’s comments on the geologic age of the study area, and we agree with the reviewer that the Zhurong landing site is within the “Late Hesperian lowland unit” based on Tanaka’s geological map of Mars (Tanaka et al., 2014). The “Late Hesperian lowland unit” represents a vast area in the northern hemisphere of Mars, which was dated using crater counting chronology on craters of large size. However, some local resurfacing processes may not have affected such large craters with elevated rims at the same time that they may have totally buried small craters. Thus, resurfacing events in more localized areas, such as around the Zhurong landing site, were not included in Tanaka’s standard geologic

map. While Tanaka's map provides excellent information regarding the geologic ages across Mars on a large scale, crater counting chronology is also always further conducted on a more refined spatial scale to discern the geological evolutionary history of such a localized area as a rover landing site.

In fact, a recent study by Wu et al. (2021) shows that resurfacing has occurred at the Zhurong landing area during the Middle Amazonian Epoch. As shown in {REDACTED} (a figure modified from Figures 2, 14, and 15 in Wu et al., 2021), Site 1 (a ~160,000 km² area including the Zhurong landing site) is located within the Late Hesperian lowland unit {REDACTED} of Tanaka's geologic map, and the yellow dots in {REDACTED} represent all the craters used to date the geologic age of Site 1 (Wu et al, 2021). {REDACTED} shows the derived absolute model ages of Site 1 using the crater size-frequency distribution (CSFD) measurements based on the craters counted in Figure {REDACTED}. The overall model age is consistent with that in Tanaka's geologic map. Also, one can clearly see that there is a kink in the CSFD curve, showing a Middle Amazonian age of ca. 1.6 Ga (fitting-crater diameter is from 0.41–1.1 km) in the plot, which indicates that a later resurfacing event occurred at Site 1 around 1.6 Ga. In addition, another recent work by Zhao et al. (2021) conducted crater counting over a much smaller area (~600 km²) including the Zhurong's landing site and their results (Figure 3 in their paper) indicate an even more recent local emplacement/resurfacing event/process that occurred in the study area in the Late Amazonian.

{REDACTED}

Most of these references were cited in the original manuscript. In the revised manuscript, we make sure to cite all these papers. We have also added a few sentences to describe the resurfacing events that occurred at the study area in the Middle to Late Amazonian (Lines 73–76), providing readers necessary background information for this critical assertion of age. Also, as you will see in response to later comments, we took more consideration of the Amazonian resurfacing in our revised interpretation of the upper fining-upward sequence in the radar profile (Lines 148–155).

References:

- Tanaka, K. L., Robbins, S., Fortezzo, C., Skinner Jr, J. & Hare, T. M. The digital global geologic map of Mars: Chronostratigraphic ages, topographic and crater morphologic characteristics, and updated resurfacing history. *Planetary and Space Science* 95, 11–24 (2014).
- Wu, X. et al. Geological characteristics of China's Tianwen-1 landing site at Utopia Planitia, Mars. *Icarus* 370, 114657 (2021).

Zhao, J. et al. Geological Characteristics and Targets of High Scientific Interest in the Zhurong Landing Region on Mars. *Geophysical Research Letters* e2021GL094903 (2021).

2. Lines 98-99, 373-374, ExFig2 I'm not familiar with techniques to suppress (or attenuate, line 99) noise, nor am I familiar with effects of doing this, so it's incumbent on the authors to describe this process fully and transparently in order to do a good evaluation. I'm worried that suppressing noise via an un-described mechanism may introduce artifacts or a different kind of noise that cannot be as easily characterized or detected. Further, in this paper readers/reviewers cannot effects of doing "noise suppression" because this step was completed simultaneously with "background removal, band-pass filtering, [and] AGC" (Data Figure 2). This is too much processing in a single step for a good evaluation, and I am unable to assess if any of these processes introduced artifacts that can alter interpretations. This is critical to address because every interpretation of the images hinges on this step.

Thanks for pointing out this detailed method. We agree that the processing workflow should be presented step by step for the readers to understand how the images (e.g., Extended Data Fig. 2) are generated. For a better exposition of the data processing, in the revision, we now show the raw data in a new Extended Data Figure 1, and have also added the radar profiles after applying self-test trace removal, trace spacing regularization, DC shift removal and time zero correction in a new Extended Data Figure 2. We also add the radar profiles after subsequently applying background removal, band-pass filtering and amplitude compensation (AGC), and random noise attenuation in the new Extended Data Figure 3 {REDACTED}. As band-pass filtering does not lead to noticeable changes in the data compared with the preceding background removal step, the result after band-pass filtering is not presented individually. We have also modified the corresponding text in the Methods in the revised manuscript (Lines 353–388).

Random noise attenuation is a data processing step widely used in seismic exploration for suppressing incoherent noise. Generally, this step is not required in the regular processing procedure of GPR data with a very shallow detection depth (about several meters) and high antenna power. For the RoPeR, the detection depth of the low-frequency channel is around 80 m. Due to the geometric diffusion and scattering of transmitted signals, the received signals in the deeper part of its range are very weak and hard to be distinguished from random noise. Thus, to enhance the signal-to-noise ratio of the low-frequency radar profile, especially for the deep part, we applied random noise attenuation to the data from the low-frequency channel using an effective method published recently. We have added text describing the technique and provided the appropriate reference in the Methods (Lines 386–388). After the application of random noise attenuation, the undesirable random noise is effectively suppressed, and useful signals are well preserved (REDACTED; Extended Data Fig. 3c in the revised manuscript). We think this step is necessary because the enhanced amplitudes in the results facilitate better geological interpretation.

{ REDACTED }

3. Minor: I'm also not familiar with the "self-test trace removal". Are these traces that are used for calibration? A citation for previous usage of this is necessary.

Thanks for your honesty and request for more clarity. The self-test traces from the RoPeR low-frequency channel are used to check the status of the RoPeR module. These traces contain no effective subsurface information, and should be excluded before processing. As the "Self_test" mode is new, and to our knowledge, first designed for the payload system of the Tianwen-1 mission, there is no published literature to date describing this mode and the processing step of self-test trace removal. As such, it cannot be referenced. Instead, in the revision, we have added a brief description of the mode and the corresponding processing step in the Methods (Lines 357–361).

4. On interpretation, the manuscript presents three plausible scenarios to explain the observations and identifies one scenario that is clearly favored (Section starting in line 150; Figure 3). But I'm worried that the data are being over-interpreted by making conclusions about ambiguous data. Assuming that no artifacts were introduced in making Figure 2, without clear reflections, it may be that no unique solution can be found, meaning that there are likely more interpretations possible.

For example, In Figure 2 80-100 m, the manuscript suggests that this low reflection zone (basal layer) is caused by a gradual decrease in permittivity ($\epsilon \sim 5$) that, but it could easily arise from

- 1) a loss of signal before 80 m, meaning it's all noise
- 2) the transition at ~ 80 m is to solid bedrock with no internal reflections, leaving only noise to interpret (similar to the idea for the moon, line 196).

Thanks for these comments. The estimates of dielectric permittivity at 80–100 m depth are not reliable because there are few effective signals for constraining dielectric permittivity. We agree that this basal layer either consists of solid bedrock with weak internal reflections or is out of the detection range of the RoPeR data. Considering the ambiguity, we do not interpret this layer. For more transparency on this limitation, we made some modifications in the text when describing the image for this lowest depth range (Lines 117–120).

As I see it, the dashed line on the far right of Figure 2 is completely unconstrained and should have a very large gray band around it, reaching to >9 .

Thanks for pointing out this problem. We have updated the dielectric permittivity plot in Figure 2c with the part below ~ 80 m being largely removed. We also now explicitly state in the revised figure caption that dielectric permittivity in this lowest depth range is not well constrained.

I think there is a possibility of over interpreting done at shallower depths too (see next comment)

5. Based on the previous comment and the methods (especially lines 414-424), I'm worried that the radar propagation modeling was carefully tuned to reproduce the observations and that once found, additional tuned solutions weren't sought. Radar attenuation via transmission losses (ϵ'') was not taken into account, so finding the signal strength at depth with this model leaves the possibility of other solutions being missed.

Thanks for your comment regarding transmission losses and our modeling. We agree that transmission losses (dielectric loss ϵ'') can influence the strength and thus the penetrating depth of radar signals. As there is little liquid water detected within the penetration depth range (0–80 m) of the RoPeR radar data (center frequency of 55 MHz) at the Zhurong landing site, the corresponding transmission losses or the loss tangent are expected to be small. As you can see from Figure { REDACTED } 3 { REDACTED } in Lai et al. (2020), previous radar simulation results with comparable frequency contents to the RoPeR low-frequency data show consistent reflection patterns in the shallow part ($<2,200$ ns or <100 m, with relatively strong reflections), but different attenuation characteristics in the deeper part, where the loss tangent (the ratio of the imaginary and real parts of permittivity) varies by nearly an order of magnitude (from 0.001–0.009). This observation suggests that small transmission losses (small values of loss tangent, i.e., on the order of 0.01 or less) appear to have a negligible influence on the reflection patterns above 100 m, although they may make it more challenging to identify deeper signals.

To further investigate the potential influence of transmission losses on the reflection pattern, we applied different loss tangent values to the synthetic radar data that were generated from a layered model with gradually increasing dielectric permittivity. Similar frequency contents to the RoPeR low-frequency data were considered in our simulation. The radar profiles of the synthetic data are shown in Figure R2a, and the corresponding results after background removal and amplitude compensation (AGC) are shown in Figure R2b. We can see that even though the amplitudes of the original data decrease significantly with depth in the lossy medium (in the case of $\tan\delta = 0.015$ in Figure R2a), the reflection pattern (0 to 1,250 ns, similar to the data from the RoPeR low-frequency channel) in the processed profile is similar to other cases with smaller transmission losses (Fig. R2b). This observation thus also indicates the relatively minor influence of transmission losses on the reflection pattern.

In our interpretation of the radar profile, we focus on the reflection pattern in the depth range where radar signals are visible. The numerical model (Extended Data Fig. 7) considered in our study is used to validate the interpretation of the imaged reflection pattern at shallow depths (0 to 80 m in Figure 2a) rather than to investigate the deeper attenuation characteristics. That is why we did not consider transmission losses in our numerical simulation. We have added these explanations to the Methods in the revised manuscript (Lines 420–427).

{ REDACTED }

Reference:

Lai, J. et al. First look by the Yutu-2 rover at the deep subsurface structure at the lunar farside. Nature communications 11, 1–9 (2020).

Fig. R2 Radar simulation results using different values of loss tangent ($\tan\delta$) for frequency contents similar to the RoPeR low-frequency data. **a**, Original data profiles with $\tan\delta = 0$, 0.005, 0.01, and 0.015 (where $\tan\delta = 0$ means lossless medium). **b**, Data profiles after applying background removal and AGC. While the amplitudes of the original synthetics in **a** are different, the reflection patterns in **b** are all similar.

Minor Comments:

6. Line 33 “southern Utopia Planitia” is a huge region. This is at one specific site

We replaced “southern Utopia Planitia” with “a southern marginal area of Utopia Planitia”.

7. Line 52 missing article “targeted the southern”

Corrected.

8. Lines 64-68 This feels like a distraction because those features are approximately 1000 km farther north, where the climate is very different from southern UP.

We agree that these features, especially the giant polygons, are more prominent farther north of Utopia, but these features have nevertheless all been identified and mapped previously in the Zhurong landing area (for example, Plate 1 in Hiesinger and Head, 2000; Figure 13 in Ivanov et al., 2014; Figure 2 in Mills et al., 2021; Figure 4 in Wu et al., 2021; Figure 6 in Ye et al., 2021; and Figure 4 in Zhao et al., 2021). In the revision, we have now cited all these recent mapping contributions. { REDACTED }

{REDACTED}

{REDACTED}

Additional references:

Hiesinger, H. & Head III, J. W. Characteristics and origin of polygonal terrain in southern Utopia Planitia, Mars: results from Mars Orbiter Laser Altimeter and Mars Orbiter Camera data. *Journal of Geophysical Research: Planets* 105, 11999–12022 (2000).

Ivanov, M. A., Hiesinger, H., Erkeling, G. & Reiss, D. Mud volcanism and morphology of impact craters in Utopia Planitia on Mars: Evidence for the ancient ocean. *Icarus* 228, 121–140 (2014).

Mills, M. M., McEwen, A. S. & Okubo, C. H. A Preliminary Regional Geomorphologic Map in Utopia Planitia of the Tianwen - 1 Zhurong Landing Region. *Geophysical Research Letters* 48, e2021GL094629 (2021).

Ye, B. et al. Geomorphologic exploration targets at the Zhurong landing site in the southern Utopia Planitia of Mars. *Earth and Planetary Science Letters* 576, 117199 (2021).

9. Lines 68-74 Again, a likely distraction since even Figure 1b shows that the volcanic flows didn't reach the Zhurong landing site,

We agree that the volcanic flows from Elysium may not reach our study area. However, the influence of the Elysium volcanism, or any unrecognized late-stage volcanism, cannot be fully ruled out due to possible later reworking. We have revised the text accordingly as follows (Lines 69–76):

“However, due to possible subsequent reworking, it is unclear whether or not the volcanic flows of the Elysium eruption or unrecognized late-stage volcanism on Mars has affected the vast plains far from Elysium Mons including the Zhurong landing area where the closest volcanic outcrops are located several hundreds of kilometers to the north (Fig. 1b). Recent geomorphological and chronological studies of the Zhurong landing site suggest that resurfacing probably occurred in this area during the Middle to Late Amazonian epochs (Wu et al., 2021, Zhao et al., 2021), but the nature of such resurfacing events has been poorly constrained.”

10. Lines 77-78 It's tempting, as the manuscript does, to call these “high-frequency” and “low-frequency” channels, and for the purpose of this manuscript it's probably fine. But there is also value in acknowledging that the higher frequency channel is fully in the “Ultra high frequency” UHF part of the spectrum (300 - 3,000 MHz). In a similar way, the “low-frequency” channel is mostly within the “Very High Frequency” VHF domain (30-300 MHz). If the authors are inclined, I'd recommend calling them the VHF channel and UHF Channel, but this is a preference and not strictly necessary.

In the reference that we cited here (ref. 14; Zhou et al., 2020), and in all other published papers about RoPeR onboard the Zhurong rover, the two channels RoPeR is equipped with were described as “high-frequency” and “low-frequency” with their specific frequency bands provided. We prefer to maintain the same descriptive precedent set by these papers.

Nonetheless, to avoid potential misunderstandings cautioned by the reviewer, we have added text to the Methods to specify these terms as they relate to the Zhurong rover (Lines 348–350).

That said, on line 97 and throughout, I recommend changing the text to say “high- and low’ frequency channels.” For example, “we analyzed data from both the high- and low-frequency channels and ...”
We have modified the text throughout the manuscript as suggested.

11. In several locations the manuscript uses strong language when there should be some hedging. For example, on line 102-103 “radar profile shows clear reflections within the depth range of 10-80 m”. These aren’t clear to me. Usually, “clear reflections” would be used to describe discrete reflectors that can be traced. Here’s an example I found on the web: <https://www.si.edu/newsdesk/photos/mars-sharad-data>. I don’t see anything like that in Figure

What I see is “regions with variable reflectivity” or some qualitative description like that. Other examples include

Line 126 “indicating a graduate change in clast size” - this is an interpretation of noisy data that does not clearly indicate clast size changes.

Line 134 “independently verifies the stratigraphic interpretation” - it’s hard to independently verify something with only one dataset. Normally, numerical models are used to “support the stratigraphic interpretation”

Line 163 “synthetic modeling results consistently indicate ...” very strong words.

Line 169 “observed” should be “interpreted”

Line 191 “strong reflections”

Line 218 “clear reflections” - again, they aren’t clear. Because water would attenuate the signal, one could change the text to read “shows a radar signal within the depth range”

We agree that more objective descriptions are more appropriate for reporting the results and have incorporated your suggestion throughout the manuscript. We address each of the reviewer’s comments with related changes made throughout the manuscript by line number:

Lines 102–103 (now lines 92–93), “clear reflections” has been changed to “depth-varying reflection characteristics”

Line 126 (now lines 113–115), In the third layer within the depth range of 30–80 m, noise has been largely suppressed (Extended Data Fig. 3), and we can extract structural information from the radar image (Fig. 2a in the main text). To be more precise, we have modified the text by replacing “indicating a graduate change in clast size” with “indicating a relatively gradual change in clast size with depth between the upper and lower parts of the layer”.

Line 134 (now line 123), “independently verifies the stratigraphic interpretation” has been replaced with “supports the stratigraphic interpretation”

Line 163, this sentence has been removed, and the entire part of the geological interpretation has been rewritten.

Line 169, this sentence has been removed.

Line 191, “strong reflections” no longer appears in the revised manuscript.

Line 218 (now line 180), “clear reflections” has been changed to “radar signals”

12. Line 147 This isn’t a diffraction survey, so I’m confused by this terminology.

In this study, we estimated the dielectric permittivity by diffraction analysis, as described in the Methods (Lines 396–412). To avoid misunderstanding, we have changed the text in the caption of Figure 2 to now read (Lines 329–331):

“The red line is the averaged 1D dielectric permittivity profile, and the bounding gray band denotes the variations around the average dielectric permittivity at each depth. Dielectric permittivity below ~80 m has not been constrained (see text for details)”.

13. Lines 158-161 Big leap in logic to get to southern UP.

By assimilating comments and suggestions on the interpretation from both reviewers, we have rewritten the entire part of geological interpretation. In the revised manuscript, please see the completely revised 4 associated paragraphs (Lines 126–174).

14. Line 211-212, Figure 3 This comes out of nowhere.

We have deleted the sentence in question. Also, we added a question mark in Figure 3c after “Deep surviving brines”, as the current study is unable to constrain whether or not there are deep surviving brines at depths >80 m.

15. Line 238-240 (last sentence): This is a strange ending. I’m not sure that the interpretations are strongly suggestive that “shallow water has fully escaped” because of the ambiguities associated with the interpretations. It may be sufficient to say that this investigation has not found evidence of water in the upper 80 m.

We have modified the text as (Lines 200–203): “Our results from southern Utopia Planitia do not provide evidence for the presence of water in the upper ~80 m. Liquid water and/or brines, if they exist, may have been buried at greater depths (Fig. 3c), mostly beyond the penetrating depth of RoPeR.”

16. Line 471, I’m not able to access the data without a login.

The Mars Rover Penetrating Radar (RoPeR) data used in this study are processed and produced by the Ground Research and Application System (GRAS) of China's Lunar and Planetary Exploration Program, and provided by China National Space Administration (CNSA) (<https://clpds.bao.ac.cn/web/enmanager/mars1>). According to the Data Release Management Measures of the Lunar Exploration and Space Engineering Center (e.g., <https://clpds.bao.ac.cn/web/enmanager/noticelist?detailId=855149>), users with data requirements must first go to the Engineering Center to apply for data. Anyone whose application is approved can access the data required. It may take some time to fully open the data to the international community.

Reviewer #2:

The manuscript provides analysis of ground penetrating radar (GPR - RoPeR Instrument) data collected within the Utopian Basin of Mars by the Tianwen-1 - Zhurong rover. Through the collection of >1km profile, RoPeR offers a unique and important insight into the geology of this fascinating region of the planet. The data processing steps are clearly outlined in the methodology section (and apply well established techniques from terrestrial GPR studies) and the final radargram (Fig 2/Extended Data Fig. 4) displays horizontal variations in subsurface scattering properties that are interpreted to be evidence of layering within the upper ~80m. Drawing on previous orbital data analysis the manuscript considers multiple possible geologic explanations for the RoPeR results. Ultimately, a layered sequence of sedimentary deposits laid down by episodic flooding is favored.

Based on the evidence presented I am not convinced that the RoPeR data supports the interpretation presented. The subsurface structures are not well defined in the GPR profile and thus a wide range of processes could be responsible for the crude layering observed. To support the flood deposit interpretation, the GPR data is compared to synthetic data generated from a model of suspended boulders within a sandy matrix (Extended data fig 8). The real data displays a more complex pattern of scattering relative to the simulated data, especially between 400 and 600 ns and thus the model doesn't appear to be adequate interpretation of actual results. I therefore suggest that more modeling is performed to test the other three scenarios considered on page 9.

Thanks for the comments. The radar profile (Figure 2 and Extended Data Figure 4, the latter of which is now Extended Data Figure 5 in the revision) does not show typical continuous reflections from layered structures, but is dominated by incoherently reflected waves from small-scale structures, such as rocky blocks of varying size. The permittivity contrasts between the rocky blocks (e.g., boulders) and the background matrix materials are responsible for the generation of the observed reflections (e.g., Zhang et al., 2021). In our synthetic tests, we do not intend to match the incoherent waveforms; instead, we try to recover the variations of reflection patterns with depth by designing models with various numbers and sizes of boulders. We mainly focus on the two aspects of the radar data—relative energy variation and depth distribution of reflections—to constrain both the

distribution of rocky blocks and average permittivity in the depth range of 0–80 m. Such kinds of synthetic experiments have been successfully applied in analyses of lunar penetrating radar data (e.g., Lai et al., 2020; Lv et al., 2020; Zhang et al., 2021) to constrain the properties of ejecta deposits on the far side of the Moon.

In the revision, we have updated the synthetic model and provided an additional way to better compare the synthetics and the observations. The synthetic model presented in the new Extended Data Figure 7 {REDACTED} generates depth variations in the reflection pattern similar to the observations, although the modeled waveforms do not provide an exact match for the real radar data (Extended Data Figures 8a and 8c [REDACTED]). We therefore also compare the corresponding normalized average strength envelopes from the synthetics and from the real radar data (Extended Data Figures 8b and 8d {REDACTED}), which highlights their similarity in the overall variation pattern with depth. This additional comparison verifies the sizes and depth distribution of rocky blocks in the Zhurong landing area, which is then considered as the observational basis for the following geological interpretation.

{REDACTED}

{REDACTED}

References:

- Lai, J. et al. First look by the Yutu-2 rover at the deep subsurface structure at the lunar farside. *Nature communications* 11, 1–9 (2020).
- Lv, W., Li, C., Song, H., Zhang, J. & Lin, Y. Comparative analysis of reflection characteristics of lunar penetrating radar data using numerical simulations. *Icarus* 350, 113896 (2020).
- Zhang, J. et al. Lunar regolith and substructure at Chang'E-4 landing site in South Pole–Aitken basin. *Nature Astronomy* 5, 25–30 (2021).

I also appreciate that the manuscript draws on published mapping efforts to support the favored interpretation, but the resolution of the map in Fig 1 is very coarse relative to the ~1km rover profile. I suggest the authors use Tianwen-1 image data as well as HiRISE and CTX images to further investigate the landing site region to provide support for the subsurface analysis. Likewise, does the surface analysis provided by the rover provide any evidence consistent with the flood deposit hypothesis?

Thanks for the excellent suggestion. We have modified Figure 1 by showing the latest local geomorphologic map of the Zhurong landing area (Mills et al., 2021), as well as adding text conveying that the local geological mapping results (using HiRISE and CTX images) provide evidence for the occurrence of younger resurfacing events at the landing area during the Middle Amazonian (Wu et al., 2021) to Late Amazonian (Zhao et al., 2021). We have also rewritten the entire part of the geological interpretation by assimilating the comments and suggestions of both reviewers. We combine both the structural features constrained by the radar data and recent geological mapping and dating results based mainly on the HiRISE and CTX images to support our interpretation. We also consider long-term weathering and repeated impacts in addition to transient floods as alternative explanations for the formation of the upper layer beneath the regolith. In the revised manuscript, please see the completely revised 4 associated paragraphs (Lines 126–174). At present, the processing and analysis of the Tianwen-1 data is still ongoing, and the preliminary results obtained so far are insufficient to support any preference(s) among the proposed scenarios. Future analyses utilizing both *in-situ* and orbital data from Tianwen-1, combined with systematic comparisons with data from other exploration missions, will provide further constraints on the origin and spatiotemporal variations of the subsurface stratification at the landing area.

Reviewer Reports on the First Revision:

Referee #1:

This is a much improved version, and I believe that all of the comments from myself and the other reviewer were addressed very well.

I still have some concerns about the ambiguity of the interpretations, which are certainly plausible but non-unique. I believe enough hedging has been introduced to acknowledge the ambiguities, and the novel nature of this dataset is a strong motivator to publish as is.

Isaac Smith

Referee #2:

The additions made to the text and figures in response to the reviews have improved the manuscript substantially.

I believe the paper is largely ready for publication. The only outstanding issue is that while flood deposits are an explanation for the radar results, they are certainly not the only cause. RoPeR detects at least one subsurface zone of enhanced scattering, but this may not necessary be caused by a boundary of suspended boulders that transition to finer grains (and even if it does, flooding is not the only explanation for such a sequence). For example voids in a buried degraded lava flow(s) or blocky material within ejecta deposits could also result in similar GPR returns. I am not advocating that the manuscript go through all possible scenarios, and in fact I feel the discussion in the main body of the text provides a more than adequate interpretation of the results. My issue is with the abstract that promotes the flooding interpretation too strongly. I suggest the text be reworded to make clear that flooding is just one of several possibilities.

Author Rebuttals to First Revision:

Reviewer #1:

This is a much improved version, and I believe that all of the comments from myself and the other reviewer were addressed very well.

I still have some concerns about the ambiguity of the interpretations, which are certainly plausible but non-unique. I believe enough hedging has been introduced to acknowledge the ambiguities, and the novel nature of this dataset is a strong motivator to publish as is.

Isaac Smith

We appreciate the positive feedback and are delighted that our revisions meet the expectations.

Reviewer #2:

The additions made to the text and figures in response to the reviews have improved the manuscript substantially.

I believe the paper is largely ready for publication. The only outstanding issue is that while flood deposits are an explanation for the radar results, they are certainly not the only cause. RoPeR detects at least one subsurface zone of enhanced scattering, but this may not necessarily be caused by a boundary of suspended boulders that transition to finer grains (and even if it does, flooding is not the only explanation for such a sequence). For example voids in a buried degraded lava flow(s) or blocky material within ejecta deposits could also result in similar GPR returns. I am not advocating that the manuscript go through all possible scenarios, and in fact I feel the discussion in the main body of the text provides a more than adequate interpretation of the results. My issue is with the abstract that promotes the flooding interpretation too strongly. I suggest the text be reworded to make clear that flooding is just one of several possibilities.

We appreciate the positive comments as well as the suggestion for the first paragraph. We have modified the sentence in question in the first paragraph to both reflect the ambiguities raised by Reviewer 2 and make our interpretation more clear at the same time, and now reads: "While alternative models deserve further scrutiny, the new radar image suggests the occurrence of episodic hydraulic flooding sedimentation that is interpreted to represent the basin infilling of Utopia Planitia during the Late Hesperian to Amazonian."